# Predicting Alcohol Consumption Patterns for Individuals with a User-Friendly Parsimonious Statistical Model

**DOI:** 10.3390/ijerph20032581

**Published:** 2023-01-31

**Authors:** Wenbin Liang, HuiJun Chih, Tanya Chikritzhs

**Affiliations:** 1School of Public Health, Fujian Medical University, Fuzhou 350108, China; 2Menzies School of Health Research, Royal Darwin Hospital Campus, Tiwi, NT 0810, Australia; 3National Drug Research Institute, Faculty of Health Sciences, Curtin University, GPO U1987, Perth, WA 6845, Australia; 4Curtin School of Population Health, Faculty of Health Sciences, Curtin University, GPO U1987, Perth, WA 6845, Australia

**Keywords:** prediction modelling, alcohol drinking, risky consumption, predictive accuracy, ROC curve

## Abstract

Many studies on the relationship between alcohol and health outcome focus primarily on average consumption over time and do not consider how heavy per-occasion drinking may influence apparent relationships. Improved methods concerning the most recent drinking occasion are essential to inform the extent of alcohol-related health problems. We aimed to develop a user-friendly and readily replicable computational model that predicts: (i) an individual’s probability of consuming alcohol ≥2, 3, 4… drinks; and (ii) the total number of days during which consumption is ≥2, 3, 4… drinks over a specified period. Data from the 2010 and 2011 National Survey on Drug Use and Health (NSDUH) were used to develop and validate the model. Predictors used in model development were age, gender, usual number of drinks consumed per day, and number of drinking days in the past 30 days. Main outcomes were number of drinks consumed on the last drinking occasion in the past 30 days, and number of days of risky levels of consumption. The area under ROC curves ranged between 0.86 and 0.91 when predicting the number of drinks consumed. Coefficients were very close to 1 for all outcomes, indicating closeness between the predicted and observed values. This straightforward modelling approach can be easily adopted by public health behavioral studies.

## 1. Introduction

Globally, alcohol is a major risk factor for mortality and morbidity [1,2,3]. Alcohol consumption increases an individual’s risk of short-term harms and social problems such as car crashes, falls and violent behavior, as well as risks of morbidity and premature mortality from a broad range of long-term physical and mental health conditions [4,5,6,7,8,9]. Short-term harms tend to occur soon after a drinking occasion as a result of the acute effects of alcohol on the brain and nervous system whereas long-term harms arise from regular and ongoing alcohol exposure [4,9,10].

Alcohol consumption is a multi-dimensional behavior. Within individuals, consumption patterns can vary from day-to-day, episode-to-episode and over the individual’s lifespan. This variability makes the accurate measurement of intake and the accounting for different patterns of consumption a difficult task, particularly when consumption is self-reported [11]. On the whole, research studies which attempt to measure individual alcohol exposure and its relationship to disease or injury could be improved by better accounting for the multiple dimensions of alcohol consumption, and, in particular, the occurrence of heavy-drinking occasions [10,12]. Many studies on the relationship between alcohol and chronic disease, for instance, focus primarily on average alcohol consumption over time (e.g., months, years) and do not consider how heavy per-occasion drinking may influence apparent relationships [10,13]. Both the rate and the amount of consumption have been found to be associated with the likelihood of blackout and hangover, with higher levels of binge drinking being associated with more negative consequences [14,15]. Regardless of whether or not individuals plan their drinking occasions, environmental factors (e.g., time spent at a bar, company at the bar) and cognitive state can alter rate of (compulsive) consumption and affective state, which may, in turn, alter subsequent reporting accuracy [14,16,17,18,19,20,21]. Moreover, the seriousness of negative consequences (e.g., hospitalization due to intoxication and/or injury) can affect the amounts consumed during subsequent drinking occasions [22]. It should not be surprising, therefore, that analyses based on average consumption estimates can lead to model misspecification and spurious conclusions about alcohol’s relationship to various harms [3,4,9,10,12]. Therefore, consumption data concerning the most recent or ‘last’ occasion, rather than only data on the maximum or total number of drinks consumed during a single occasion within a fixed period (e.g., last month, year), are required to account for random error in analytical models. Improved methods and analyses of the distribution of alcohol use are essential for informing the extent of alcohol-related death and disability, particularly when consumption of varied amounts could contribute to different types of alcohol-related health problems [2,12,23].

The aim of this study was to develop a user-friendly and readily replicable computational model that predicts an individual’s alcohol consumption pattern based on self-reported responses to a minimum set of items commonly included in alcohol-use surveys. We set out to predict the probability of alcohol consumption above a given level (i.e., ≥2, 3, 4… drinks) during a drinking occasion or a drinking day and the total number of days during which an individual consumes alcohol above a given level (i.e., number of days with 2+, 3+, 4+… drinks) over a specified period (i.e., monthly, yearly).

## 2. Materials and Methods

We postulated that a drinking occasion may be thought of as a simple mechanistic process involving a series of ‘mini drinking episodes’ during which alcohol has a specific probability of being consumed. For example, while attending a social outing (an ‘occasion’), Drinker X may begin drinking upon arrival and drink more or less continuously throughout the entire occasion. Drinker Y, however, may consume alcohol while awaiting a meal (a ‘mini episode’), drink only water (or nothing) during the meal, and then consume alcohol again when dessert arrives or after dessert is consumed (another ‘mini episode’). Alcohol may, therefore, be consumed continuously (i.e., without ending a single mini episode), but there is also an equal probability that the mini episode may cease before consuming the next drink, or even before finishing the first drink.

We further postulated that the total amount of alcohol consumed during a mini episode can be modelled using a geometric distribution that counts the number of failures of independent trials (i.e., the number of drinks consumed without stopping) before the first success (i.e., stopping drinking and ending a mini episode). The amount of alcohol consumed during the entire occasion (e.g., a drinking day), may be calculated by summing the quantity of alcohol consumed in each mini episode. An individual’s drinking occasion may, therefore, be described using negative binomial distribution, given that a negative binomial distributed variable can be viewed as the sum of some independent and identically distributed (i.i.d.) geometrical variables (i.e., the amount of alcohol consumed in each mini episode). This study developed and tested a model to describe and predict individual alcohol consumption patterns based on the negative binomial distribution.

### 2.1. Dataset and Key Variables

We used datasets from the 2010 and 2011 National Survey on Drug Use and Health (NSDUH) https://www.samhsa.gov/data/data-we-collect/nsduh-national-survey-drug-use-and-health (accessed on 6 May 2020) to develop and validate our statistical model. Briefly, NSDUH surveys are multistage representative national samples of non-institutionalized populations aged 12 years or older living in the United States. Details of the survey method have been described previously [24,25]. This study included participants who consumed at least one drink in the past 30 days and provided answers to all alcohol questions.

As our intention was that the model be as widely applicable as possible, when selecting predictor variables, we drew on only a limited set of four basic measures that can be readily derived from questions asked in most national surveys. The four predictor variables, indicated elsewhere as key drinking norms [20], were (i) age (recoded from 17 to 9 categories: 12–14; 15–17; 18–19; 20–25; 26–29; 30–34; 35–49; 50–64; and 65+; years old), (ii) gender, (iii) usual number of drinks consumed per day, and (iv) number of drinking days in the past 30 days. The alcohol questions used to measure the usual number of drinks per day and the number of drinking days in the past 30 days were as follows: (a) “*On the days that you drank during the past 30 days, how many drinks did you usually have? Count as a drink a can or bottle of beer; a wine cooler or a glass of wine, champagne, or sherry; a shot of liquor or a mixed drink or cocktail.*”, and (b) “*Think specifically about the past 30 days, up to and including today. During the past 30 days, on how many days did you drink one or more drinks of an alcoholic beverage?*”.

The key outcome variable was number of drinks consumed on the last drinking occasion in the past 30 days. This variable was used to both develop and validate the model and was derived from the commonly asked survey question: “*Earlier, the computer recorded that you drank at least one alcoholic beverage within the past 30 days, please think about the last time you drank any alcoholic beverage. How many drinks did you have that time?*”. Data on consumption during the ‘last’ occasion were found to be essential to minimize reporting errors resulted from environmental and individual factors, as well as the consequences of risky drinking occasions [14,18,19,20,21,22].

We also used two further questions regarding the number of days when risky levels of use were exceeded to validate the model. A variable for all participants that counted the number of days in which they had had 5+ drinks on the same occasion in the past 30 days was created from the commonly asked survey question “*During the past 30 days, on how many days did you have 5 or more drinks on the same occasion? By “occasion,” we mean at the same time or within a couple of hours of each other*.”, The question asked in the NSDUH is also asked in the Australian National Health Survey. The NSDUH also asked females (only) “*During the past 30 days, on how many days did you have 4 or more drinks on the same occasion?*” from which we created a variable for number of days in which they had had 4+ drinks on the same occasion in the past 30 days for females only.

The usual number of drinks per day and the number of drinks consumed on the last drinking occasion had upper bounds of 24 and 20 drinks, respectively. If, for example, a participant reported having had 30 drinks on the occasion when they last drank, the data would be recoded to 20 drinks. Although the upper bounds might not capture the very high-level episodic-drinking populations, these upper bounds were chosen because they approximated the 99th percentile of the variable plus 1 unit of its standard deviation and were accounted for by the model (see Section 2.2 Model specification).

Before conducting analyses, the two years of survey data were combined into a single dataset. Approximately 75% of participants were randomly selected from the full dataset to form the model training sample (*n* = 34,389). The remaining participants formed the testing sample (*n* = 11,637). The training sample was used to estimate the model parameters, and the testing sample was used to validate the model.

### 2.2. Model Specification and Estimation

Let Y denote the amount of alcohol that may be consumed in a day, whereas y_j_ denotes the observed number of drinks consumed on the last drinking occasion in the past 30 days by the jth participant; r denote the number of mini episodes; p denote the probability of success (stop drinking); P(Y = y|r, p) denote the probability mass function that specifies the probability of observing data vector y given the parameter vectors r and p. Since this study only includes those who had at least one drink within the past 30 days (Y ≥ 1), the model uses P(Y = y|r, p, Y ≥ 1), which is the probability mass function given r, p and Y ≥ 1. Y is bounded by 20 drinks; hence, the model uses P(Y ≥ 20|r, p, Y ≥ 1) as the sum of the probability mass function of all y ≥ 20 given r, p and Y ≥ 1.

Therefore, as illustrated in Appendix A, the log likelihood function, lnl_j_, is:$$\mathrm{lnL}=\overset{n}{\underset{j=1}{\sum }}{\mathrm{lnl}}_{j}$$
and the full model is:$$a_{p}+x_{\mathrm{pj}}b_{p}=a_{p}+x_{3j}b_{p3}+x_{4j}b_{p4}+x_{5j}b_{p5}+x_{6j}b_{p6}+x_{7j}b_{p7}$$

The full model was fitted using the training sample with maximum likelihood estimation (mL) in Stata [26]. All non-significant terms were then removed and the model re-run. The process of removing non-significant terms and re-running the model was repeated until all remaining terms were significant. Model coefficients from the final model were then applied to calculate r and p for both the training sample and the testing sample.

### 2.3. Model Validation

The final model was validated by measuring the predictive accuracy on three different outcomes in the testing data: (i) number of drinks consumed on the last drinking occasion in the past 30 days; (ii) number of days in which they had had 5+ drinks on the same occasion in the past 30 days; and (iii) number of days in which they had had 4+ drinks on the same occasion in the past 30 days for females. Given the calculated r and p, the probability of consuming any number y (≥1) and above of drinks on an occasion can be estimated with the following:$$P\left(Y\geq y\;|\;r,p,\;Y\geq 1\right)=\frac{1-\sum_{y=0}^{\left(y-1\right)}P\left(Y=y\;|\;r,p\right)}{1-P\left(Y=0\;|\;r,p\right)}$$

The number of days when 5+ drinks were consumed on the same occasion during the past 30 days was considered to be a variable with binomial distribution with n trials (n equals the number of drinking days in the past 30 days) and probability of success equal to P(Y ≥ 5|r, p, Y ≥ 1). Thus, the predicted number of days with 5+ drinks was directly estimated using P(Y ≥ 5|r, p, Y ≥ 1) × the number of drinking days in the past 30 days. The same method was used to estimate the number of days when 4+ drinks were consumed on the same occasion in the past 30 days.

The variable for number of drinks consumed on the last drinking occasion was recoded into 11 binary variables G_i_ using i = 2, 3, 4, …, 12 drinks as cut-off points. For example, the binary variable G_2_ (i.e., 2 drinks) = 0 if Y < 2 and G_2_ = 1 if Y ≥ 2 drinks. For each binary outcome variable G_i_, individual probabilities that G_i_ = 1 were calculated for both the training and testing samples, and the probabilities converted into a natural logarithm. Logistic regression was fitted using the natural logarithm of the calculated probabilities from the training sample; the area under the receiver operating characteristic (AU-ROC) curve was then estimated using the postestimation command in Stata for the training sample and the testing sample.

The performance of the predictive model on binary outcomes was assessed using the AU-ROC curve, and model performance for predicting the continuous outcomes was assessed based on goodness of fit (R^2^) [27,28]. The ability of the model to predict the number of days when 5+ drinks were consumed on the same occasion during the past 30 days (and 4+ for females) was tested by applying slope-only simple linear regression models to the training and testing samples and comparing the observed number of days to the estimated number of days. A coefficient close to 1 and a high R^2^ value indicated that the predicted value closely approximated the observed value.

## 3. Results

### 3.1. Predictabilty and Validation of the Model

The ability of the model to predict the three tested outcomes was excellent. When predicting the binary-transformed variables for number of drinks consumed on the last drinking occasion, AU-ROC curves ranged between 0.86 and 0.91 for the testing sample (Table 1).

As shown in Table 2, using the testing sample, R^2^ values for models that predicted the number of days when 5+ drinks were consumed on the same occasion in the past 30 days for both genders or 4+ drinks on the same occasion for females were 0.79 and 0.74, respectively, and the coefficients were very close to 1 for both outcomes. This indicates that the values predicted based on the model coefficients were close to the observed values in the testing sample.

Model coefficients that determined r and p are shown in Table 3. Application of these values in the model is illustrated in Section 3.2.

### 3.2. Application of the Model

We can apply the specific coefficients of the model (Table 3) to calculate mini episodes (r) and probability of stopping drinking (p) for individuals. For instance, a female aged between 35 and 49 years, who usually consumes three drinks per drinking day, and usually consumes alcohol on 20 days in a month, can have her mini episodes (r) and probability of stopping drinking (p) estimated as:$$\begin{array}{c} r_{j}=1+\exp\left(a_{r}+x_{\mathrm{rj}}b_{r}\right)\\ =1+\exp(2.239-0.081-0.079-3.109\times \ln\left(\frac{1}{3}\right)-0.943\times {\left(\ln\left(\frac{1}{3}\right)\right)}^{2}-2.055\times \ln\left(20\right)+0.368\times \ln{\left(20\right)}^{2}\\ =5.49 \end{array}$$
$$\begin{array}{c} p_{j}=\frac{\exp\left(a_{p}+x_{\mathrm{pj}}b_{p}\right)}{1+\exp\left(a_{p}+x_{\mathrm{pj}}b_{p}\right)}\\ =\frac{\exp\left(3.077-0.964\times \ln\left(\frac{1}{3}\right)\right)-0.563\times \left(\ln\left(\frac{1}{3}\right){)}^{2}-1.925\times \ln\left(20\right)+0.325\times \ln{\left(20\right)}^{2}\right)}{1+\exp\left(3.077-0.964\times \ln\left(\frac{1}{3}\right)\right)-0.563\times \left(\ln\left(\frac{1}{3}\right){)}^{2}-1.925\times \ln\left(20\right)+0.325\times \ln{\left(20\right)}^{2}\right)}\\ =0.648 \end{array}$$

We can also estimate an individual’s probability of alcohol consumption at or above any level (up to 20 drinks) on one occasion. For instance, for a drinker who has two mini drinking episodes (r = 2) and a probability of stopping drinking of 50% (p = 0.5), the probability of consuming two drinks on one occasion (e.g., a drinking day), is P(Y = 2|r = 2, p = 0.5)/P(Y ≥ 1|r = 2, p = 0.5) = 0.25. For the same drinker, the probability of consuming three drinks is P(Y = 3|r = 2, p = 0.5)/P(Y ≥ 1|r = 2, p = 0.5) = 0.167, and the probability of consuming four drinks is P(Y = 4|r = 2, p = 0.5)/P(Y ≥ 1|r = 2, p = 0.5) = 0.104.

Note that calculating P(Y = 2|r = 2, p = 0.5) only requires application of the negative binomial probability mass function, while calculating P(Y ≥ 1|r = 2, p = 0.5) requires application of the negative binomial complementary cumulative distribution function, i.e., one minus the negative binomial cumulative distribution function in statistical software, such as Stata^®^, SPSS^®^ or R^®^.

For instance, the probability of the individual described above consuming 2+ drinks on a drinking day is P(Y ≥ 2|r = 2, p = 0.5)/P(Y ≥ 1|r = 2, p = 0.5) = 0.667, where both P(Y ≥ 2|r = 2, p = 0.5) and P(Y ≥ 1|r = 2, p = 0.5) can be obtained using the negative binomial complementary cumulative distribution function. Similarly, for 3+ and 4+ drinks, the probabilities are P(Y ≥ 3|r = 2, p = 0.5)/P(Y ≥ 1|r = 2, p = 0.5) = 0.417; and P(Y ≥ 4|r = 2, p = 0.5)/P(Y ≥ 1| r = 2, p = 0.5) = 0.25, respectively.

Estimation of the total number of days during which an individual consumes alcohol above a given level over a specified period is then straightforward. For instance, following the assumptions of the previous example, and if that individual has 10 drinking days a month, the regularity of them consuming 2+; 3+; or 4+ drinks in a month can be estimated as, on average, 0.667 × 10 = 6.67; 0.417 × 10 = 4.17; and 0.25 × 10 = 2.5 days, respectively.

## 4. Discussion

In this study, we developed and tested a modelling approach to describe and predict drinking patterns for individuals. We considered that in order to be of practical utility, the model itself should be parsimonious and the input data readily obtainable from items commonly used in basic alcohol-use surveys. We demonstrated that our model provides good predictive accuracy on all of the three outcomes applied in the validation exercise, namely (i) number of drinks consumed on the last drinking occasion in the past 30 days; (ii) number of days in which they had had 5+ drinks on the same occasion in the past 30 days; and (iii) number of days in which they had had 4+ drinks on the same occasion in the past 30 days for females. Our model only requires basic demographic information (age and gender) and two simple questions on alcohol consumption: usual number of drinks consumed per day and number of drinking days in the past 30 days. Similar questions are commonly used in alcohol-consumption surveys. Indeed, using our approach, the estimation of an individual’s probability of alcohol consumption at or above any level (up to 20 drinks) on one occasion or frequency of alcohol consumption at or above any level for a period of time is valid and straightforward. A simple conversion table and/or automated calculator (similar to that illustrated in Section 3.2) could be constructed to assist researchers and clinicians in determining a patient’s likelihood of drinking cessation, likelihood of heavy consumption, and periodicity of heavy use. With improved understanding of patient drinking patterns and risk severity through the use of our user-friendly and readily replicable survey, clinicians will be better equipped to deliver appropriate and effective interventions in a timely manner, such as reducing drinking during weekend nights for young adults aged 18–29 years old [22], encouraging pregnant women to abstain from alcohol [29], providing brief motivational interventions to reduce alcohol consumption among non-dependent patients [30], in order to improve patient functioning and reduce alcohol-related physical and mental health problems [31].

We used self-reported information on the usual number of drinks per day and the frequency of drinking in the past 30 days to underpin estimates of quantity and frequency of alcohol consumption for individuals. Based on the hypothesis that a drinking occasion is a mechanistic process, these two variables can be used to identify functions of the unobserved number of mini episodes (the parameter r) and the probability of success/stopping drinking (p). Usual number of drinks on drinking days provides a good approximation of the average number of drinks per drinking occasion, and for a negative binomial distributed variable Y~ NBin(r, p) the mean of Y equals to: r/p × (1 − p), while the proportion of drinking days in the past 30 days (drinking frequency) provides an approximation for P(Y ≥ 1|r, p). The postulated mechanistic process of drinking occasions is well-supported by the empirical data used in this study. Nevertheless, these findings require independent verification by other analysts drawing on survey data from a wide range of populations.

### Limitations and Future Perspectives

The model described here was developed from a large random sample of the United States’ general population and it is likely that the coefficients will vary in other populations. It remains necessary that the model be grounded in basic survey data derived from the population of interest rather than relying on the generalizability of coefficients from other populations. For instance, our model may not be generalizable to people with alcohol-use disorders or those who consume more than the upper bounds applied in our models (i.e., more than 24 drinks per day or more than 20 drinks consumed on the last drinking occasion). Nevertheless, adjustment for specific populations is feasible through the adaptation of our modelling approach.

It is likely that the performance of the model may be improved by including more variables, and this may be a preferred approach for analysts wishing to develop a more complex model for a specific population. However, considering the complexities that environmental and individual factors bear on alcohol consumption [14,18,19,21], increasing the number of variables is likely to increase the magnitude of reporting errors and biases. This may, in turn, affect ability to estimate true individual-specific distributions of alcohol consumption using self-reported surveys. Given that the current model needs only minimal information about individuals and does not require a specific unit of measurement for alcohol (e.g., standard drinks), we anticipate that this approach will be widely applicable to populations and studies that collect a minimum level of alcohol consumption data but lack reliable measurement on drinking patterns and occasional drinking in particular. We also anticipate that this approach can be applied in other research areas, such as binge eating.

In prioritizing the development of a user-friendly and readily replicable computational model, we did not consider the varying issues of self-reported data (e.g., under-reporting, recall bias, sampling issue, etc.) in our models. Future analyses may adjust for known magnitude of underreporting if higher accuracy than those reported by our models (i.e., AU-ROC > 0.9) was preferred. Analysts may prefer to fit more variables in alternative models while balancing model accuracy and practicality. Future studies may explore the generalizability of our model using data collected from countries and sub-populations with different drinking norms.

## 5. Conclusions

We have described and tested a computational model that predicts alcohol consumption patterns for individuals, including occasional drinking, with good predictive accuracy using age, gender and only two self-reported alcohol survey items. This straightforward modelling approach may be readily adopted and applied in alcohol studies worldwide, with potential application in other substance-use and addiction studies.

## Figures and Tables

**Table 1 ijerph-20-02581-t001:** Ability of the model to predict alcohol consumption * above a given level (≥2, 3, 4, …, 12 drinks) during the last drinking occasion.

Cut-Off Points	Areas Under the ROC Curve	
	Training Dataset	Testing Dataset
2	0.857	0.862
3	0.880	0.882
4	0.893	0.896
5	0.897	0.898
6	0.901	0.902
7	0.905	0.904
8	0.909	0.909
9	0.908	0.905
10	0.910	0.908
11	0.914	0.911
12	0.914	0.911

* outcome = 1 if number of drinks ≥ cut-off point; outcome = 0 otherwise.

**Table 2 ijerph-20-02581-t002:** Ability of the model * to predict number of days when an individual consumes 5+ drinks (both sex) or 4+ drinks (female only) in the past 30 days.

	R-Squared	Coefficient	95% CI	
number of days with 5+ drinks	0.796	0.987	0.978	0.996
number of days with 4+ drinks	0.742	1.051	1.035	1.067

* assessed on the testing sample using linear regression models.

**Table 3 ijerph-20-02581-t003:** Estimated model coefficients for calculation of (i) number of mini episodes (r) and (ii) probability of stopping drinking (p).

Coefficients for Number of Mini Episodes (r)	Coefficient	95% CI	
Age group			
12–14	−0.033	−0.126	0.061
15–17	0.014	−0.017	0.046
18–19	0.020	−0.009	0.048
20–25	reference		
26–29	−0.026	−0.061	0.010
30–34	−0.014	−0.050	0.022
35–49	−0.081	−0.109	−0.053
50–64	−0.215	−0.265	−0.165
65+	−0.423	−0.527	−0.319
Gender			
Male	reference		
Female	−0.079	−0.097	−0.060
Ln(1/K *)	−3.109	−3.269	−2.950
(Ln(1/K *))2	−0.943	−1.005	−0.882
Ln(M #)	−2.055	−2.273	−1.838
(Ln(M #))2	0.368	0.314	0.423
Constant	2.239	2.000	2.478
Ln(1/K *)	−3.109	−3.269	−2.950
(Ln(1/K *))2	−0.943	−1.005	−0.882
Coefficients for probability of stopping drinking (p)			
Ln(1/K *)	−0.964	−1.092	−0.836
(Ln(1/K *))2	−0.563	−0.614	−0.512
Ln(M #)	−1.925	−2.132	−1.719
(Ln(M #))2	0.325	0.274	0.375
Constant	3.077	2.871	3.283
Ln(1/K *)	−0.964	−1.092	−0.836

* K: usual reported number of drinks consumed per day; # M: reported number of drinking days in the past 30 days.

## Data Availability

No new data were created or analyzed in this study. Data sharing is not applicable to this article.

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
