# Peer review of "Predicting Alcohol Consumption Patterns for Individuals with a User-Friendly Parsimonious Statistical Model"

_ijerph, 2023, doi:10.3390/ijerph20032581_

Round 1
Reviewer 1 Report
The article titled Predicting Alcohol Consumption Patterns for Individuals with a User-friendly Parsimonious Statistical Model is scholarly piece of work that takes on a pressing health issue – alcohol consumption. This study put forth a statistical technique that has high accuracy in predicting alcohol use behaviors, which is of use to alcohol use researchers and suits the scope of the journal. The article was well-written in English and easily comprehensible. Method was described in detailed steps and results were clearly presented. However, I have a couple of minor concerns. First, the authors did not cite the relevant literature on statistical modeling using count variables, which assumes a negative binomial distribution. Given that the focus of the study lies in developing a statistical model properly handing the count variables (i.e., alcohol consumption), failure in citing the statistical literature undermines the rationale and validity of the study. Further, authors need to include discussions on how sampling weights were applied to the analytical sample given the multi-stage study design. This would inform other researchers in appropriately analyzing the national datasets with complex sampling designs. experimental methods. Last, authors might consider submitting the relevant STATA code as supplemental materials that might be of use to other applied researchers in conducting alcohol research.
Reviewer 2 Report
I believe the methods you used to build the model are appropriate. However, I have a few issues with the manuscript:
(1) INTRODUCTION, page 1, line 32: Please use the term "crashes" instead of "accidents". Accidents have the connotation of no causes.
(2) In using the two questions from the survey, isn't it crucial to get the number of drinks consumed over a time period? It is quite different for someone to have 5 drinks over 2 hours vs someone who has 5 drinks over 5 hours! How do you account for those differences?
(3) DISCUSSION, page 8, line 281: You say "...clinicians will be better equipped to deliver appropriate and effective interventions..." Please give citations here and give examples of how your model will actually accomplish that. Two or three examples.
(4) Is it possible to test your model on real people drinking under normal conditions and being breath tested to determine the accuracy of your model?
(5) CONCLUSIONS, page 9, line 331: You say can be "applied in alcohol studies..." Again, citations and examples, please.
The main question addressed by the research was the ability to predict alcohol consumption of individuals based upon their responses to two questions.
The topic addresses a specific gap but it is not clear how it will be used.
It adds an easy way to estimate alcohol consumption of individuals.
The authors should consider testing the model in the real world.
The conclusions need relevant examples of how the model can be used in the treatment of alcohol abuse.
The references lack citations on how predicting alcohol consumption can help in treatment.
